# ENSEMBLE SYSTEMS REPRESENTATION FOR FUNCTION LEARNING OVER MANIFOLDS

## ABSTRACT

Function learning concerns with the search for functional relationships among datasets. It coincides with the formulations of various learning problems, particularly supervised learning problems, and serves as the prototype for many learning models, e.g., neural networks and kernel machines. In this paper, we propose a novel framework to tackle function learning tasks from the perspective of *ensemble systems and control theory*. Our central idea is to generate function learning algorithms by using flows of continuous-time ensemble systems defined on infinite-dimensional Riemannian manifolds. This immediately gives rise to the notion of *natural gradient flow* that enables the generated algorithms to tackle function learning tasks over manifolds. Moreover, we rigorously investigate the relationship between the convergence of the generated algorithms and the dynamics of the ensemble systems with and without an external forcing or control input. We show that by turning the penalty strengths into control inputs, the algorithms are able to converge to any function over the manifold, regardless of the initial guesses, providing *ensemble controllability* of the systems. In addition to the theoretical investigation, concrete examples are also provided to demonstrate the high efficiency and excellent generalizability of these "continuous-time" algorithms compared with classical "discrete-time" algorithms.

## 1 INTRODUCTION

The core challenge in numerous scientific and engineering disciplines revolves around the learning of nonlinear functions through the utilization of parametric or nonparametric models. This is frequently achieved by minimizing error functions across continuous, high-dimensional spaces. Such function learning (FL) tasks are centered on the representation of relationships between input and output data, employing continuous variable functions. For example, in supervised learning, regression analysis is usually formulated as the search of functions in a predetermined hypothesis space of candidate functions that most closely fit the data (Abu-Mostafa et al., 2012; Goodfellow et al., 2016b), and classification aims at learning functions (*the decision boundary*) taking values on finite sets consisting of the class labels that map each datapoint to its class. On the other hand, many learning techniques and tools also include FL as an essential step, e.g., reinforcement learning (RL). Notably, one of the major uses of neural networks is to approximate functions as indicated by the universal approximation theorem, and kernel machines are designed to parameterize feature maps, those are, functions that transform raw data to feature vectors. Following the broad scope, FL is prevalent across diverse domains of science and engineering, ranging from system identification (Schoukens & Ljung, 2019; Ljung, 1999; Narendra & Kannan, 1990) and RL (Sutton & Barto, 1998; Bertsekas et al., 2000; Doya, 2000) to model learning and inverse problems (Tarantola, 2005).

The fundamental concept behind FL is the generation of a sequence of function estimates, starting from an initial approximation, that progressively approach the target function by leveraging observation data. The primary challenges in efficiently addressing this problem encompass the formulation of an effective update (learning) rule and the selection of an appropriate initial approximation to ensure the convergence of the sequence towards the target function. Furthermore, when the learning process involves searching for functions defined on or constrained to manifolds, it introduces an additional layer of complexity, particularly when gradient-based techniques are employed (see Figure 1 (b)).

Conversely, the problem of steering nonlinear functions that take values on manifolds has been extensively explored within the field of *ensemble dynamics and control theory*. This intricate problem finds applications in various domains such as quantum control, neuromodulation, swarm robotics, among others (Li et al., 2011). The main objective of this paper is to leverage the advancements in ensemble dynamics and control theory to unlock fresh perspectives for tackling FL problems and to address some of the associated challenges.

## 2 RELATED WORKS

Here we review relevant works in ensemble systems (ESs) and control theory, learning and dynamics, and Riemannian optimization techniques. **ESs and Control.** This work is primarily built up on ensemble control theory, which is related to the study of populations of dynamical systems (DSs). Fundamental investigations

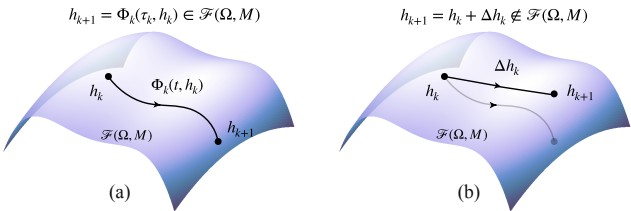

Figure 1: Comparison between the proposed ensemble systems-theoretic FL and classical function leaning. In particular, for a FL task over a manifold, the learning sequence generated by the proposed algorithm is guaranteed to stay on the manifold (left), while the one generated by classical learning algorithms may leave the domain manifold (right).

into ensemble DS and their properties, e.g., controllability and observability, have been conducted in series of works over the last two decades (Li, 2006; 2011; Zeng et al., 2016; Chen, 2020; Narayanan et al., 2020). Specifically, ensemble controllability conditions have been widely studied for both time-varying and time-invariant linear (Li, 2011; Li et al., 2020; Zeng & Allgöewer, 2016), bilinear (Li & Khaneja, 2009; Zhang & Li, 2021), and some classes of nonlinear ESs (Kuritz et al., 2018; Li et al., 2013).

**Learning and DS.** Understanding learning algorithms from the DSs perspective started from the use of strong solutions to stochastic ordinary differential equations driven by Brownian motions to approximate stochastic recursive algorithms (Borkar & Mitter, 1999). More recently, the connection between control systems and certain classes of computational neural networks have been studied in (Weinan, 2017; Haber & Ruthotto, 2017; Lu et al., 2018; He et al., 2016). In particular, these developments view the common learning problems, such as weight identifiability from data (Albertini & Sontag, 1993), controllability (Sontag & Sussmann, 1997; Sontag & Qiao, 1999), and stability (Michel et al., 1989; Hirsch, 1989) of neural networks, from a DS viewpoint. In this context, function approximation problems and the concept of *universality* of a class of deep residual networks were analyzed through the lens of homogeneous ensemble DS, which relies on initial conditions of the ensemble (Tabuada & Gharesifard, 2020; Agrachev & Sarychev, 2020). Different from the works presented earlier (Tabuada & Gharesifard, 2020; Agrachev & Sarychev, 2020), we introduce a novel concept that interprets the evolution of an iterative FL algorithm as the temporal propagation of an inhomogeneous ensemble DS. The attainment of convergence towards a desired function is viewed as the resolution of a steering problem inherent to this system. To be more precise, we demonstrate that the dynamic characteristics of the learning algorithm can be scrutinized and understood through a dynamically equivalent system—an inhomogeneous ensemble control system—that exhibits reduced sensitivity to initial conditions.

**Natural Gradients and Riemannian Optimization.** The concept of 'natural gradient' has facilitated the exploration of stochastic gradient methods on statistical manifolds using information geometry techniques (ichi Amari, 1998; Martens, 2020). In addition, for addressing challenges in semi-supervised learning, robot learning, and optimization on finite-dimensional manifolds, Riemannian geometry techniques have been proposed. This paper further develops these concepts within the ensemble DS framework, expanding the notion of natural gradients to encompass infinite-dimensional Riemannian manifolds (RMs) for FL based on manifolds.

**Our Contributions.** We focus on the fundamental question on how to appropriately generate FL algorithms by using DS evolving on RMs. Namely, can we formally represent an iterative algorithm as the time evolution of a continuous-time DS? If so, what is the relationship between the convergence of the algorithm and the dynamics of the system, and what are the necessary and sufficient conditions to guarantee the algorithm convergence in terms of the dynamics-related properties of the system? We shall answer these highly non-trivial questions by making the following contributions.

• We introduce the notion of *inhomogeneous ensemble DS* defined on infinite dimensional RMs, and then use their flows to generate FL algorithms, giving rise to a dynamic FL framework.

• We characterize the convergence of FL algorithms in terms of stability and controllability of ESs. In particular, we show that FL algorithms generated by controllable ES have the global convergence property and are guaranteed to converge, regardless of the initial guesses.

• We integrate geometric information into ESs for FL problems over manifolds by expanding the idea of *natural gradients* (ichi Amari, 1998) to develop the notion of *natural gradient flow*. Specifically, it guarantees that the generated sequences of functions stay on the manifolds where the functions learning problems are defined, as shown in Figure 1.

In addition, we provide examples to demonstrate the applicability as well as the advantages of the proposed ensemble-system theoretic FL framework. In particular, we observe high efficiency (i.e., faster convergence) and excellent generalizability via incorporation of early stopping.

## 3 PRELIMINARIES: CONTROL OF ESs

*An ES* is a parameterized family of DS defined on a manifold $M \subseteq \mathbb{R}^n$ of the form

$$\frac{d}{dt}x(t, \beta) = f(x(t, \beta), \beta, u(t)), \tag{1}$$

where the system parameter $\beta$ takes values on $\Omega \subseteq \mathbb{R}^d$, $u(t) \in \mathbb{R}^m$ is the control input, and $f(\cdot, \beta, u(t))$ is a vector field on $M$ for each fixed $\beta \in \Omega$ and control input $u$. A canonical *ensemble control* task is to design a $\beta$-independent control input $u(t)$ that steers the whole family of systems from an initial profile $x_0(\beta) = x(0, \beta)$ to a desired final profile $x_F(\beta)$ for all $\beta$. Indeed, the ensemble state $x(t, \beta)$ is a function of $\beta$ defined on $\Omega$ so that the ES in equation 1 is a DS evolving on a space $\mathcal{F}(\Omega, M)$ of $M$-valued functions defined on $\Omega$.

### 3.1 ENSEMBLE CONTROLLABILITY

Controllability is one of the most fundamental properties of a DS, which characterizes the ability of the control input to precisely steer a control system between any two points (states) in the state-space. For an ES equation 1, the parameter space $\Omega$ is generally an infinite set so that the state-space $\mathcal{F}(\Omega, M)$ is an infinite-dimensional manifold; or, in another words, the system is an infinite-dimensional system. For such a system, the aforementioned classical notion of controllability is generally too restrictive (Triggiani, 1977). Hence, we introduce the concept of ensemble controllability to characterize the ability to control an ES in the approximation sense.

**Definition 1 (Ensemble controllability)** *The system in equation 1 is said to be* ensemble controllable *on the function space $\mathcal{F}(\Omega, M)$ if for any $\varepsilon > 0$ and starting with any initial function $x_0 \in \mathcal{F}(\Omega, M)$, there exist a time $T > 0$ and a control law $u : [0, T] \to \mathbb{R}^m$ that steers the system into an $\varepsilon$-neighborhood of a desired target function $x_F \in \mathcal{F}(\Omega, M)$, i.e., $d(x(T, \cdot), x_F(\cdot)) < \varepsilon$, where $d : \mathcal{F}(\Omega, M) \times \mathcal{F}(\Omega, M) \to \mathbb{R}$ is a metric on $\mathcal{F}(\Omega, M)$.*

Definition 1 shows that ensemble controllability is a notion of approximate controllability, in which the final time $T$ may depend on the approximation accuracy $\varepsilon$. Moreover, the distance function $d$ is generally assumed to be induced by a Riemannian structure on $\mathcal{F}(\Omega, M)$ (See Appendix A.4).

**Remark 1 (Ensemble controllability and function convergence)** *Ensemble controllability further conveys the idea of convergence of functions: because $\varepsilon > 0$ is arbitrary, it is necessary that $x(T, \cdot) \to x_F(\cdot)$ as $T \to \infty$. This is essentially a continuous-time analogue to the convergence of a sequence of functions, e.g., generated by a learning algorithm.*

## 4 FL FROM AN ENSEMBLE CONTROL VIEWPOINT

In this section, we develop a universal framework to transform the design of FL algorithms to that of ensemble control laws, and rigorously establish the equivalence between convergence of learning algorithms and ensemble controllability.

### 4.1 ESs ADAPTED TO FL ALGORITHMS

The widely-used approach to learning a function is to generate a sequence of functions converging to it. By treating the index of the sequence as time, it is natural to assume that the process of generating the sequence follows the time-evolution of some DS. In addition, because each term in the sequence is a function, the associated DS is necessarily an ES.

By using the same notation as in the previous section, given a function $h \in \mathcal{F}(\Omega, M)$ to be learned from the initial guess $h_0 \in \mathcal{F}(\Omega, M)$, the canonical way is to generate an iterative algorithm in the form of $h_{k+1} = h_k + \Delta h_k$, $k \in \mathbb{N}$ such that $d(h_k, h) \to 0$ as $k \to \infty$, where $\mathbb{N}$ denote the set of nonnegative integers and $\Delta h_k$ is the update rule at the $k^{\text{th}}$ iteration, generally depending on the gradient of $h_k$. To bridge the function algorithm and an ES, we think about the case that the update rule is generated by a flow on $\mathcal{F}(\Omega, M)$, instead of the function $\Delta h_k$, as

$$h_{k+1} = \Phi_k(\tau_k, h_k) \tag{2}$$

for each $k \in \mathbb{N}$. Of course, there are generally many choices of such flows leading to the convergence of the sequence $h_k$ to $h$, and how to select the best ones depends on the functional learning problem and will be the focus of the next section by using the technique of natural gradient. For now, we just pick those that are sufficiently smooth, actually continuously differentiable is already enough, so that each $\Phi_k$ is the flow of a vector field $f_k$, possibly only locally defined, on $\mathcal{F}(\Omega, M)$. We then smoothly concatenate these vector fields together, e.g., by using a partition of unity (Lang, 1999), yielding a globally defined vector field $f$ on $\mathcal{F}(\Omega, M)$, whose flow $\Phi$ necessarily satisfies $h_{k+1} = \Phi(\tau_k, h_k)$ for all $k \in \mathbb{N}$, equivalently, $h_{k+1} = \Phi(t_k, h_0)$ with $t_k = \sum_{i=0}^{k} \tau_k$. This further implies that the solution of the (unforced) ES

$$\frac{d}{dt} x(t, \beta) = f(x(t, \beta), \beta) \tag{3}$$

with the initial condition $x(0, \cdot) = h_0(\cdot)$ satisfies $x(t_k, \cdot) = h_{k+1}(\cdot)$. In this case, we say that the ES in equation 3 is *adapted to* the FL algorithm in equation 2.

As shown in the following proposition, convergence of FL algorithms generated by flows as in equation 2 can be evaluated by stability of the adapted ESs as in equation 3.

**Proposition 1** *If the sequence of functions $\{h_k\}_{k \in \mathbb{N}}$ in $\mathcal{F}(\Omega, M)$ generated by the learning algorithm in equation 2 converges to a function $h \in \mathcal{F}(\Omega, M)$, then there is an ES in the form of equation 3 defined on $\mathcal{F}(\Omega, M)$ adapted to this learning algorithm such that $h$ is an equilibrium of the system.*

*Proof.* See Appendix A.1 □

Note that Proposition 1 only demonstrates the existence of an ES being able to stabilize at the limit point of the sequence generated by the learning algorithm, and it by no means indicates that every ES adapted to the same algorithm has this property.

**Remark 2 (FL on manifolds)** *Taking the ES in equation 3 as an example, as a system defined on $\mathcal{F}(\Omega, M)$, $f$ is a vector field on $\mathcal{F}(\Omega, M)$ so that the flow, equivalently, the entire sequence of functions generated by the adapted learning algorithm in equation 2, always evolves on the manifold $\mathcal{F}(\Omega, M)$. However, the classical learning algorithm $h_{k+1} = h_k + \Delta h_k$ may result in $h_{k+1} \notin \mathcal{F}(\Omega, M)$, even with $h_k, \Delta h_k \in \mathcal{F}(\Omega, M)$, because the manifold $\mathcal{F}(\Omega, M)$ is generally not closed under the vector space operation "+", which is also illustrated in Fig. 1.*

### 4.2 DYNAMIC FL VIA ENSEMBLE CONTROL

Having established the association of stable ESs to convergent FL algorithms, in this section, we generate FL algorithms by using ESs. According to Proposition 1, it is necessary that the ES has an equilibrium point at the function to be learned. However, this is not sufficient to guarantee the convergence of the learning algorithm generated by the ES to the desired function. Additionally, the initial guess should be accurate enough in the sense of lying in the region of attraction of the equilibrium point. These conditions together then give rise to the following converse of Proposition 1.

**Proposition 2** *Consider an ES defined on the function space $\mathcal{F}(\Omega, M)$ equation 3. If $h \in \mathcal{F}(\Omega, M)$ is an equilibrium point of the system and $h_0 \in \mathcal{F}(\Omega, M)$ is in the region of attraction of $h$, then there is a FL algorithm generated by the ES which converges to $h$.*

*Proof.* The proof directly follows from the definition of equilibrium points of DS. □

Propositions 1 and 2 give a necessary and sufficient condition for convergence of FL algorithms in terms of stability of the adapted ESs. The requirement for the adapted ESs to have stable equilibrium points at the desired functions imposes strong restrictions on the system dynamics. On the other hand, the need for the initial guesses to be in the regions of attraction of the equilibrium points may lead to sensitivity of the learning algorithms generated by these ESs to the initial guesses. To waive these requirements, it is inevitable to force such ESs by external control inputs.

In the presence of a control input $u(t)$, e.g., as the ensemble control system in equation 1, the FL algorithm in equation 2 generated by the ES, more specifically the updating rule $\Phi_k(\tau_k, h_k)$, also depends on $u(t)$. As a result, it is possible to design an appropriate $u(t)$ to enforce the convergence of learning algorithm to the desired function $h$, even though $h$ may not be an equilibrium point of the uncontrolled system.

**Theorem 1** *Given an ensemble control system defined on the function space $\mathcal{F}(\Omega, M)$ equation 1, for any $h \in \mathcal{F}(\Omega, M)$, there is a FL algorithm generated by the ES converging to $h$ regardless of the initial guess if and only if the system is ensemble controllable on $\mathcal{F}(\Omega, M)$.*

*Proof.* The idea is to interpret the concept of ensemble controllability in terms of convergence of functions as motivated in Remark 1. See Appendix A.2 for details. □

Conceptually, Theorem 1 demonstrates the potential for a novel FL algorithm design method using ensemble control theory.

**Corollary 1** *For any function $h \in \mathcal{F}(\Omega, M)$, if the initial guess $h_0 \in \mathcal{F}(\Omega, M)$ of $h$ is in the controllable submanifold of the ES in equation 1 containing $h$, then there is a FL algorithm as in equation 2 generated by the ES, converging to $h$.*

*Proof.* Because any ES is ensemble controllable on its controllable submanifold, the proof directly follows from Theorem 1 by restricting the ES in equation 1 to the controllable submanifold containing $h$ and $h_0$. □

**Remark 3 (Robustness to initial guesses)** *Theorem 1 and Corollary 1 presented a distinctive feature of the FL algorithms generated by ensemble control systems, that is, the robustness to initial guesses. With the ability to manipulate the "algorithm dynamics" using a control input, initial guesses are no longer required to be close to the desired function. In particular, under the condition of ensemble controllability, the learning algorithm converges globally; otherwise, it is sufficient to set the initial guess on the same controllable submanifold as the target function.*

On the other hand, Theorem 1 and Corollary 1 also indicate that the FL algorithm design problem can be formulated as an ensemble control problem, which can be tackled by various well-developed methods, such as pseudospectral (Li et al., 2011) and iterative linearization methods (Wang & Li, 2018; Zeng, 2019).

## 5 NATURAL GRADIENT FLOW FOR GEOMETRIC FL

The concept of natural gradient was introduced for the study of stochastic gradient methods on statistical manifolds by leveraging information geometry techniques (ichi Amari, 1998). In this section, we integrate this idea with ESs to study FL problems over nonlinear manifolds.

### 5.1 NATURAL GRADIENT FLOW SYSTEM

Conceptually, the natural gradient of a real-valued function defined on a RM is a vector field characterizing the steepest ascent direction of the function, which generalizes the notion of gradient in classical multivariant calculus. Adopting the same notation as in the previous sections, for a FL problem over the, possibly infinite-dimensional, RM $\mathcal{F}(\Omega, M)$, it usually associates with a nonnegative loss function $L : \mathcal{F}(\Omega, M) \to \mathbb{R}$, and our task is to search the space $\mathcal{F}(\Omega, M)$ for a function minimizing $L$. To find the steepest decent direction, we recall that the differential $dL$ of $L$ is a 1-form on $\mathcal{F}(\Omega, M)$ such that $dL(p) \cdot v$ gives the directional (Gateaux) derivative of $L$ along the

direction of $v$ for any $p \in \mathcal{F}(\Omega, M)$ and $v \in T_p\mathcal{F}(\Omega, M)$, the tangent space of $\mathcal{F}(\Omega, M)$ at $p$ (Lang, 1999). As a RM, $T_p\mathcal{F}(\Omega, M)$ is Hilbert space equipped with an inner product $\langle \cdot, \cdot \rangle$, the restriction of the Riemannian metric on $T_p\mathcal{F}(\Omega, M)$. Then, the *Resize representation theorem* can be applied to identify $dL(p)$ with an element in $T_p\mathcal{F}(\Omega, M)$ (Folland, 2013), denoted by $\operatorname{grad} L(p)$. By varying $p$, we then obtain a vector field $\operatorname{grad} L$ on $\mathcal{F}(\Omega, M)$, which is called the *natural gradient* of $L$. By construction, the natural gradient satisfies $\langle \operatorname{grad} L, V \rangle = dL \cdot V$ for any vector field $V$ on $\mathcal{F}(\Omega, M)$, which then guarantees that $-\operatorname{grad} L$ gives the steep decent direction of $L$ at every point in $\mathcal{F}(\Omega, M)$ (ichi Amari, 1998). Then, recall the FL algorithm equation 2, the best choice of the flow $\Phi$ will be the flow of the vector field $-\operatorname{grad} L$, and the corresponding ES $\frac{d}{dt}x(t, \beta) = -\operatorname{grad} L(x(t, \beta))$ is named as the *natural gradient flow* system.

In practice, it is common to minimize $L$ under some penalties $R_i : \mathcal{F}(\Omega, M) \to \mathbb{R}$ with $i = 1, \ldots, m$, e.g., for improving the generalizability of the learning algorithm (Goodfellow et al., 2016a). In this case, we also involve these penalty functions into the natural gradient flow system and make it a control-affine ES defined on $\mathcal{F}(\Omega, M)$ as

$$\frac{d}{dt}x(t, \beta) = -\operatorname{grad} L(x(t, \beta)) + \sum_{i=1}^{m} u_i(t)\operatorname{grad} R_i(x(t, \beta)), \tag{4}$$

where the control inputs $u_i$ play the role of time-varying penalty coefficients. Note that because the natural gradient vector fields $\operatorname{grad} L(x(t, \beta))$ and $\operatorname{grad} R_i(x(t, \beta))$ are defined through the Riemannian metric of $\mathcal{F}(\Omega, M)$, the natural gradient flow system in equation 4 also documents rich geometric information of $\mathcal{F}(\Omega, M)$, which then generates geometry-preserving FL algorithms.

**Remark 4** *The most interesting and counter-intuitive part in the natural gradient flow system in equation 4 is the role of the control vector fields played by the regulators or exploratory signals. Contrary to regular learning algorithms in which these terms combat with loss functions, resulting in some sacrifice for algorithm performance, our results reveal that they, serving as control vector fields, tend to make the natural gradient flow system ensemble controllable, which in turn leads to global convergence of the generated algorithm. Geometrically, with these penalties, the system can be steered to, equivalently, the generated algorithm can learn, more functions (any functions if controllable), in addition to those along the natural gradient direction of the cost function.*

## 5.2 DYNAMIC FL FOR PARAMETERIZED MODELS

In practice, to learn a function $h : \Omega \to M$, it is highly inefficient, or even impractical, to search for the entire space of $M$-valued functions defined on $\Omega$. Fortunately, with some prior knowledge about $h$, it is possible to focus the learning on a subspace $\mathcal{F}$ of this function space. Of particular interest, it is common to consider the case in which functions in $\mathcal{F}$ can be indexed by parameters taking values in a set $\Theta$. Formally, this means there is a bijective map $\iota : \Theta \to \mathcal{F}$, which without loss of generality can be assumed to be a diffeomorphism by giving $\Theta$ a smooth structure compatible with the one on $\mathcal{F}$. Consequently, the FL problem can be formulated as the search of an element $\theta \in \Theta$ such that the function indexed by $\theta$ best approximates $h$ in the sense of the loss function $\bar{L} : \Theta \to \mathbb{R}$, given by $\bar{L} = L \circ \iota$, being minimized at $\theta$. Moreover, $\Theta$ can be made a RM as well by pulling back the Riemannian metric on $\mathcal{F}$ through $\iota$. To be more specific, given any vector fields $V$ and $W$ on $\Theta$, we define the pull back metric $\langle \cdot, \cdot \rangle_\Theta$ on $\Theta$ by $\langle V, W \rangle_\Theta = \langle d\iota \cdot V, d\iota \cdot W \rangle_\mathcal{F}$, where $\langle \cdot, \cdot \rangle_\mathcal{F}$ denotes the Riemannian metric on $\mathcal{F}$. This then enables the calculation of the natural gradient vector field of $\bar{L}$, and the natural gradient flow system in equation 5 becomes a control-affine system on $\Theta$ as

$$\frac{d}{dt}\theta(t) = -\operatorname{grad} \bar{L}(\theta(t)) + \sum_{i=1}^{m} u_i(t)\operatorname{grad} \bar{R}_i(\theta(t)), \tag{5}$$

where $\bar{R}_i = R \circ \iota$ for the penalties $R_i : \mathcal{F} \to \mathbb{R}$, and the natural gradient $\operatorname{grad}$ is with respect to the Riemannian structure on $\Theta$. Actually, by using the pull back metric, $\Theta$ is essentially another copy of $\mathcal{F}$ as a RM so that the system in 5 is just the original system in equation 4 in the $\theta$-coordinates. Therefore, all the discussion about ESs and the generated learning algorithms remains valid for the system in equation 5.

## 6 EXAMPLES AND DISCUSSIONS

**Dynamic Curve Fitting.** The first example is a curve fitting, and the purpose is to find a function having the best fit to an input-output dataset. Let $X = \{x_1, \ldots, x_N\} \subset \mathbb{R}^n$ and $Y = \{y_1, \ldots, y_N\} \subset \mathbb{R}$ be the input and output data, respectively, and $\mathcal{F}$ denote some space containing functions from $\mathbb{R}^n$ to $\mathbb{R}$, then the curve fitting problem can be formulated as $\min_{h \in \mathcal{F}} L(h)$ for some loss function $L : \mathcal{F} \to \mathbb{R}$. In general, $\mathcal{F}$ is chosen to be a finite-dimensional vector space with a given a basis $\{\varphi_i\}_{i=0}^{r-1}$ and the quadrantic loss $L(\theta) = \frac{1}{2} \sum_{i=1}^{N} |y_i - \sum_{j=0}^{r-1} \theta_j \varphi_j(x_i)|^2$ over $\theta = (\theta_0, \ldots, \theta_{r-1})'$. In this case, the natural gradient reduces to the usual gradient, given by, $\operatorname{grad} L(\theta) = H'H\theta - H'Y$, where $H \in \mathbb{R}^{N \times r}$ is the regressor matrix with the $(i, j)$-entry defined by $H_{ij} = \Phi_j(x_i)$ and $Y = (y_1, \ldots, y_N)' \in \mathbb{R}^N$ is the data vector. This leads to the natural gradient flow system as

$$\frac{d}{dt}\theta(t) = -H'H\theta(t) + H'Y, \tag{6}$$

Note that the system in equation 6 is a linear system whose solution is given by the variation of constant formula $\theta(t) = e^{-tH'H}\theta(0) + \int_0^t e^{(s-t)H'H}H'Y\,dt$, where $\theta(0)$ is the initial guess (Brockett, 2015). In general, the regressor matrix $H$ is full rank (unless there are redundant data) so that $-H'H$ is negative-definite, and hence $e^{-tH'H} \to 0$ as $t \to 0$. This implies that the solution of the natural gradient flow system in equation 6 converges to the solution of the regression problem regardless of the initial guess. Moreover, the invertibility of $H'H$ gives a more concrete representation of the solution $\theta(t) = e^{-tH'H}\theta(0) + (I - e^{-tH'H})(H'H)^{-1}H'Y$, where $I \in \mathbb{R}^{r \times r}$ denotes the identity matrix. When $t \to \infty$, $\theta(t) \to (H'H)^{-1}H'Y$, which exactly coincides the solution $\theta^*$ of the linear regression problem, theoretically verifying the proposed ES-theoretic approach to FL.

To demonstrate the applicability of this novel approach, we would like to learn the nonlinear function $h : [-1, 1] \to \mathbb{R}, x \mapsto \cos(1.15\pi x) + \sin(1.15\pi x)$ by using polynomial functions up to order 4, i.e., the function space $\mathcal{F}$ is the 5-dimensional vector space spanned by $\varphi_i(x) = x^i$ for $i = 0, 1, \ldots, 4$. To this end, we draw 20 samples $x_1, \ldots, x_{20}$ from the uniform distribution on [-1,1] as the input data, then perturb the values of $h$ evaluated at these points by a 0 mean and 0.05 variance Gaussian noise $\delta$ as the output data $y_i = h(x_i) + \delta$, $i = 1, \ldots, 20$. We then solve the natural gradient flow system in equation 6 numerically for $t \in [0, 100]$. The $\ell^2$-error between $\theta(t)$ and $\theta^*$ and the cost with respect to time are shown in Figure 2a, which rapidly converge to 0 and the minimum cost, respectively. Moreover, in Figure 2b, we show the polynomials with coefficients $\theta(t)$ for $t = 10, 20, \ldots, 100$, which clearly converge to the least square solution $h^*$ of the regression problem.

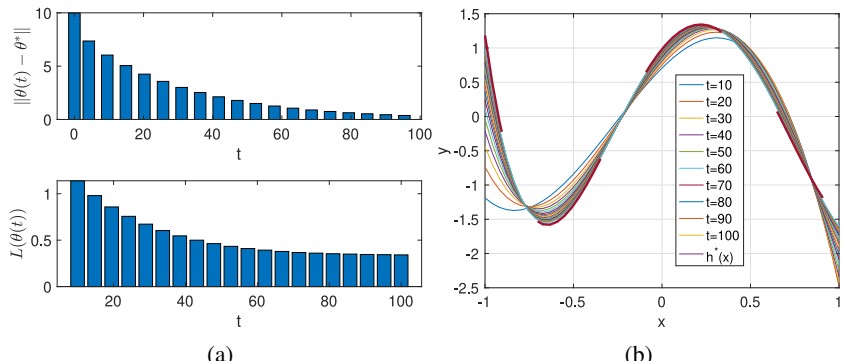

(a)         (b)

Figure 2: Dynamic curve fitting by using the natural gradient flow system in equation 6 associated with $\min_{\theta \in \mathbb{R}^4} \sum_{i=1}^{20} |y_i - \sum_{j=0}^{4} \theta_i x_i^j|^2$, and the initial condition of the system is chosen randomly from $(0, 1)$. In particular, (a) shows the time evolutions of the $\ell^2$-error between $\theta(t)$ and $\theta^*$ (top) as well as the cost $L(\theta(t))$ (bottom), where $\theta(t)$ is the solution of the system and $\theta^*$ is the least squares solution, and (b) illustrates the convergence of the polynomial functions (the solid curves) with coefficients $\theta(t)$, $t = 10, 20, \ldots, 100$ to the least squares solution (the dashed curve).

Now, if there is a penalty function, say $R(\theta) = \sum_{i=1}^{n} \theta_i$, then the loss function becomes $L(\theta) = \frac{1}{2}(Y - H\theta)'(Y - H\theta) + \lambda b'\theta$, where $b$ is the column vector whose entries are all 1. Of course, in this case, the minimum of $L(\theta)$ is not the least squares solution anymore. However, if $\lambda$ can be adjusted,

then the natural gradient flow system becomes a control system as

$$\frac{d}{dt}\theta(t) = -H'H\theta(t) + H'Y - bu(t). \tag{7}$$

By using the same dataset as above, we can check that the control system in equation 7 is controllable because the controllability matrix $W = [-b \mid H'Hb \mid -(H'H)^2b \mid (H'H)^3b \mid -(H'H)^4b]$ is full rank (Brockett, 2015). This says that for any desired final state, particularly the least squares solution $\theta^*$, there is a control input (time-varying penalty parameter) steering the system in equation 7 to $\theta^*$ regardless of the initial condition. Actually, in this case, the steering can be exact in finite time, not only in the asymptotic sense, since the system is finite-dimensional. In particular, given a final time $T$, we can systematically construct such a control input, e.g., $u(t) = -b'e^{H'Ht}W^{-1}(0,T)\xi$, where $W(0,T) = \int_0^T e^{H'Hs}bb'e^{H'Hs}ds$ and $\xi = e^{H'HT}\theta^* - \int_0^T e^{H'Hs}H'Yds - \theta(0)$. Note that this choice of $u(t)$ is actually the minimum energy control satisfying $u(t) = \mathrm{argmin}_{v(t)} \int_0^T v^2(t)dt$ over the space of all control inputs steering the system in equation 7 from $\theta(0)$ to $\theta^*$ (Brockett, 2015). The simulation results are shown in Figure 3, in which the $\ell^2$-error between $\theta(t)$ and and $\theta^*$ and the cost $L(\theta(t))$ are shown in Figure 3a, and the minimum energy control input is shown in 3b.

**Remark 5 (High efficiency of control-based FL)** *A comparison between Figures 2 and 3 reveals that the controlled natural gradient flow system in equation 7 is steered to $\theta^*$ in 20 units of time, while it costs the unforced system in equation 6 5 times of this duration, 100 units of time. This sheds light on the high efficiency of tackling learning tasks by using control techniques. Of course, the learning time can be further shortened, but larger amplitude of the control input should be expected.*

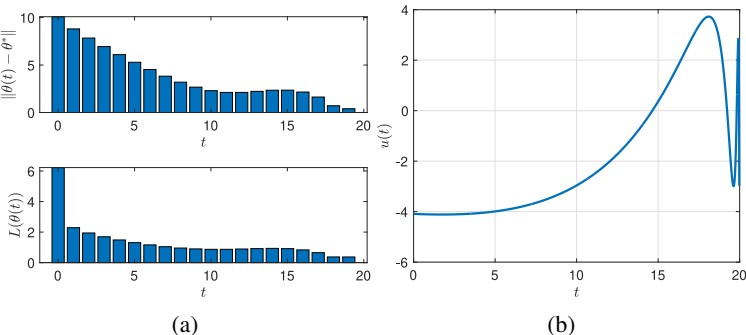

|  |  |
|:---:|:---:|
| (a) | (b) |

Figure 3: Dynamic curve fitting with a penalty function by using the controlled natural gradient flow system in equation 7 associated with $\min_{\theta \in \mathbb{R}^4} \sum_{i=1}^{20} |y_i - \sum_{j=0}^4 \theta_i x_i^j|^2$. In particular, (a) shows the time evolutions of the $\ell^2$-error between $\theta(t)$ and $\theta^*$ (top) as well as the cost $L(\theta(t))$ (bottom), where $\theta(t)$ is the trajectory of the system steered by the minimum energy control shown in (b) and $\theta^*$ is the least squares solution.

**FL over Spaces of Discrete Probability Distributions.** Leaning functions taking values on a space of discrete probability distributions has various applications, e.g., it is essential to classification problems. Geometrically, it can be shown that the space of discrete probability distributions, say on a set of $n$ elements, can be embedded into the $(n-1)$-dimensional unit sphere $\mathbb{S}^{n-1} = \{x \in \mathbb{R}^n : x'x = 1\}$ with the Riemannian metric given by the pullback of the Euclidean inner product on $\mathbb{R}^n$. More concretely, at any point $x \in \mathbb{S}^{n-1}$, every tangent vector of $\mathbb{S}^{n-1}$ is of the form $a\Omega x$ for some $n$-by-$n$ skew-symmetric matrix $\Omega$ and real number $a$ so that the pull-back metric satisfies $\langle a_1\Omega_1 x, a_2\Omega_2 x \rangle = a_1 a_2 x'\Omega_1'\Omega_2 x$ for any $a_i\Omega_i x \in T_x\mathbb{S}^{n-1}$. Suppose that we would like to learn a $\mathbb{S}^{n-1}$-valued function $h$ defined on a compact interval $K$ that minimizes the $L^2$-loss $\int_K d(h(\beta), x_F(\beta))d\beta$ for some $x_F : K \to \mathbb{S}^{n-1}$ under some penalties $R_i = \int_K \|\beta\Omega_i h(\beta)\|d\beta$, $i = 1, \ldots, m$, where $d(\cdot, \cdot)$ and $\|\cdot\|$ are the distance function and norm induced by the Riemannian metric on $\mathbb{S}^{n-1}$. Then, the natural gradient flow system is constructed as

$$\frac{d}{dt}x(t,\beta) = \beta\Big[\sum_{i=1}^m u_i(t)\Omega_i\Big]x(t,\beta), \tag{8}$$

and the learning problem can be formulated as a control problem of steering $x(t,\cdot)$ to $x_F$. To illuminate how this gradient flow system in equation 8 works, we consider the case $n = 3$, $m = 2$

with $\Omega_1 = \begin{bmatrix} 0 & 0 & -1 \\ 0 & 0 & 0 \\ 1 & 0 & 0 \end{bmatrix}$ and $\Omega_2 = \begin{bmatrix} 0 & 0 & 0 \\ 0 & 0 & 1 \\ 0 & -1 & 0 \end{bmatrix}$, $\beta \in [0.6, 1.4]$, and $x_F = (1, 0, 0)'$ the constant function. We pick the initial condition to be the constant function $x_0 = (0, 0, 1)'$, and apply the Fourier series-based method to design the control inputs (See Appendix A.3 for the detail), which are shown in Figure 4b. Steered by these control inputs, the final states, as functions of $\beta$, of the gradient flow system in equation 8 are shown in Figure 4a, with $L^2$-error $\left( \int_{0.6}^{1.4} d(x(T, \beta), x_F(\beta)) d\beta \right)^{1/2} = 0.78$. Moreover, Figure 4c shows that the system trajectory will never leave the unit sphere $\mathbb{S}^2$, equivalently, the space of discrete probability distributions on a 3-element set, as desired. It is also worth mentioning that the gradient flow ES in equation 8 is called the Bloch system, named after the Swiss-American physicist Felix Bloch, which describes the dynamics of a sample of nuclear spins immersed in a static magnetic field. Control of the Bloch system plays the fundamental role in nuclear magnetic resonance experiments, with wide-range applications including compound structure analysis, medical imaging, and quantum computing (Li & Khaneja, 2009; Li et al., 2011; Zhang & Li, 2015; Zhang et al., 2019).

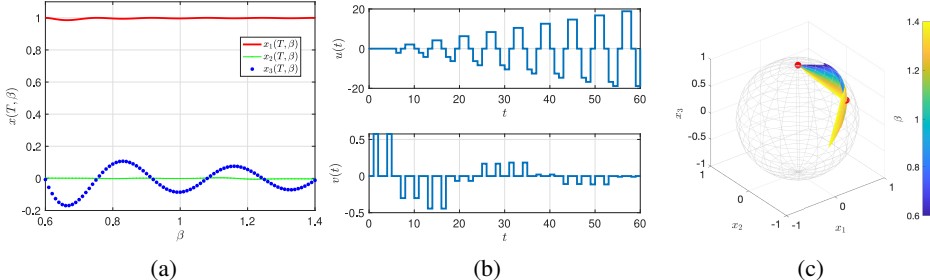

|     |     |     |
| --- | --- | --- |
| (a) | (b) | (c) |

Figure 4: FL over the space of discrete probability distributions with the natural gradient flow system in equation 8, where the task is to learn the $x_F = (1, 0, 0)'$ from the initial guess $x_0 = (0, 0, 1)'$. In particular, (a) shows the learned function by steering the system using the control inputs in (b), and (c) shows the trajectory of the natural gradient system always on the sphere $\mathbb{S}^2$ with the red dots denoting the boundary conditions.

**Remark 6 (Dynamic FL and early stopping)** *It is well-known in the machine learning society that early stopping is one of the most effective ways to improve the generalizability of learning algorithms. In the proposed ES-theoretic FL approach, early stopping can be realized by choosing a relatively small final time for the ES-adapted to a learning algorithm. In addition, compared with the classical "discrete-time" learning algorithms, the stopping criterion for this "continuous-time" algorithm does not restrict to integer time, which demonstrates a great potential to reach better generalizability.*

**Remark 7 (Natural gradient flow and stochastic gradient decent)** *Recall that the main idea of ensemble control, in addition to control of functions, is to coordinate a large population of DS by using a common control input. Hence, this control input is guaranteed to work for any sub-population of the ensemble. To further elaborate this from a learning perspective, e.g., by using the natural gradient flow system in equation 4, the algorithm generated by this system is consistent with the stochastic gradient decent.*

## 7 CONCLUSIONS

In this paper, we propose a novel FL framework through the lens of ensemble control theory, with the focus on FL problems over infinite-dimensional manifolds. The core idea is to generate a learning algorithm by using the flow of an continuous-time ES. We further rigorously investigate the relationship between the algorithm convergence and dynamics of the ES. In particular, we show that the algorithm always converges to the equilibrium points of the system, and moreover, providing the system is ensemble controllable, the generated algorithm can be guaranteed to converge to any function regardless of the initial guess. One major advantage gained from the continuous-time nature of the generated algorithm is the extraordinary ability to learn functions taking values on RMs, in which case the geometric properties of the manifolds are integrated into the ES in terms of the natural gradient flow. Moreover, examples are provide to demonstrate the high efficiency and excellent generalizability of the alogorithm. **Limitations.** Due to the nature of FL problems, the developed framework requires data consisting both of the values of a function and the corresponding preimages in the domain. This implies learning algorithms generated by ESs only works for supervised learning tasks, which indicates a limitation of this work.

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
