# A APPENDIX

## A.1 PROOF OF PROPOSITION 1

Given the FL algorithm in equation 2, the construction of the ES in equation 3 yields the existence. To make $h$ an equilibrium point of the ES, we first note that $h_k \to h$ implies that $\{h_k\}_{k \in \mathbb{N}}$ is a Cauchy sequence and hence $d(h_{k+1}, h_k) \to 0$ as $k \to \infty$. Now, we choose the flows $\Phi_k$ in the learning algorithm equation 2 in the way that $\tau_k \to 0$ and $d(h_{k+1}, h_k) \sim o(\tau_k)$, e.g., $\tau_k = \sqrt{d(h_{k+1}, h_k)}$. Recall that $h_k = \Phi(t_k, h_0)$ with $\Phi$ denoting the flow of the vector field $f$ on $\mathcal{F}(\Omega, M)$ governing the dynamics of the ES, we have

$$
\begin{aligned}
|f(h)| = \left| f\Big(\Phi\big(\lim_{k \to \infty} t_k, h_0\big)\Big) \right| &= \lim_{k \to \infty} |f(\Phi(t_k, h_0))| \\
&\leq \limsup_{k \to \infty} \frac{d\big(\Phi(t_k, h_0), \Phi(t_{k-1}, h_0)\big)}{t_k - t_{k-1}} = \limsup_{k \to \infty} \frac{d\big(h_{k+1}, h_k\big)}{\tau_k} = 0,
\end{aligned}
$$

indicating that $h$ is an equilibrium point of the system as desired.

## A.2 PROOF OF THEOREM 1

Necessity: Suppose that the system in equation 1 is ensemble controllable on $\mathcal{F}(\Omega, M)$, then for any $\varepsilon > 0$ and any initial condition $x_0 \in \mathcal{F}(\Omega, M)$, there is a control input $u(t)$ steering the system to a function $x(T, \cdot) \in \mathcal{F}(\Omega, M)$ in a finite time $T$ such that $d(x_T - h) < \varepsilon$. Then, we design a FL algorithm with the initial guess $h_0 = x_0$ converging to $h$ since $\varepsilon$ is arbitrary.

Sufficiency: Given arbitrary $h$ and $h_0$ in $\mathcal{F}(\Omega, M)$, suppose that there is a FL algorithm as in equation 2, generated by the ES in equation 1 driven by a control input $u(t)$, converging to $h$ with the initial guess $h_0$. Then, following the same proof as Proposition 1, we can show the ES stabilizes to $h$, i.e., for any $\varepsilon > 0$, there is a finite time $T$ such that the control input $u(t)$ steers the ES to $x_T$ satisfying $\|x_T - h\| < \varepsilon$. Because $\varepsilon$ is arbitrary, it concludes ensemble controllability of the system.

## A.3 FOURIER SERIES-BASED CONTROL DESIGN METHOD

Note that because the state of the Bloch (ensemble) system in equation 8 is on the unit sphere $\mathbb{S}^2$, the evolution of the system is actually given by a rotation in $\mathbb{R}^3$. The main idea of the Fourier series-based control design method for the Bloch ensemble is to decompose a desired rotation into a sequence of small angle rotations in the Fourier series manner (Zhang & Li, 2015; Zhang et al., 2019). To be more specific, applying constant control input $u(t) = u_0$ and $v(t) = v_0$ for a time duration $t_0$ results in the evolutions $\exp(\beta u_0 t_0 \Omega_y)$ and $\exp(\beta v_0 t_0 \Omega_x)$, which are rotations around $y$- and $x$-axes by the $\beta$-dependent angles $\beta u_0 t_0$ and $\beta v_0 t_0$, respectively. Consequently, the sequences of constant control inputs $(v_0, u_0, -u_0)$ and $(-v_0, u_0, u_0)$ can be applied to generate

$$
u_{1k} = \exp(-\beta \lambda_k \Omega_x) \exp(\beta c_k \Omega_y / 2) \exp(\beta \lambda_k \Omega_x)
$$

and

$$
u_{2k} = \exp(\beta \lambda_k \Omega_x) \exp(\beta c_k \Omega_y / 2) \exp(-\beta \lambda_k \Omega_x)
$$

by choosing $u_0$, $v_0$, and $t_0$ in the way that $\lambda_k = v_0 t_0$ and $c_k = 2u_0 t_0$ for $k = 1, 2, \ldots$ Then, the Baker-Campbell-Hausdorff formula implies

$$
U_{1k} = \exp\{\beta c_k [\Omega_y \cos(\lambda_k \beta) - \Omega_z \sin(\lambda_k \beta)]/2\}
$$

and

$$
U_{2k} = \exp\{\beta c_k [\Omega_y \cos(\lambda_k \beta) + \Omega_z \sin(\lambda_k \beta)]/2\}.
$$

Provided that $\beta c_k$ is small enough, we obtain

$$
U_k = U_{2k} U_{1k} \approx \exp\left[\beta c_k \cos(\lambda_k \beta) \Omega_y\right],
$$

leading to

$$
U = \prod_k U_k = \exp\left[\beta \sum_k c_k \cos(\lambda_k \beta) \Omega_y\right],
$$

which is a rotation around $y$-axis by the angle $\beta \sum_k c_k \cos(\lambda_k \beta) \Omega_y$. In particular, if we choose $\lambda_k = k$, then the rotation angle is the Fourier series expansion of a function $\theta(\beta)$. Specifically, in Section 6, we choose $\theta(\beta) = \pi/2$, the constant function $\pi/2$ on $[0.6, 1.4]$.

As mentioned in Section 6 as well, the ensemble Bloch equation in equation 8 describes the dynamics of a sample of spin-$1/2$ nuclei immersed in a static magnetic field, in which the control inputs $u(t)$ and $v(t)$ denote the external radio frequency (rf) fields applied along $y$- and $x$-axes, respectively. The dispersion parameter $\varepsilon$ is called the *rf inhomogeneity*, caused by the fact that different spins at different positions in the sample receive different strengths of the rf fields. It has been shown by practical nuclear magnetic resonance (NMR) experiments that rf fields at the ends of the coil can be as low as 60% of the rf fields at the center of the coil, corresponding to the rf inhomogeneity $\beta \in [0.6, 1.4]$.

### A.4 BASICS OF RIEMANNIAN GEOMETRY

This appendix devotes to a brief review of Riemannian geometry in the most general setting, which is also inclusive for the infinite-dimensional case.

Let $\mathcal{H}$ be a Hilbert space, a topological space $X$ is said to be a *topological manifold* modeled on $\mathcal{H}$, if $X$ is Hausdorff, second countable, and locally homeomorphic to $\mathcal{H}$, that is, every point in $X$ has an open neighborhood homeomorphic to $\mathcal{H}$. Let $\{U_\alpha\}_{\alpha \in A}$ be an open cover of $X$ such that there is a homeomoprhism $\varphi_\alpha : U_\alpha \to \mathcal{H}$ for each $\alpha \in A$, then each pair $(U_\alpha, \varphi_\alpha)$ is called a *coordinate chart* on $X$ and the entire collection of charts $\{(U_\alpha, \varphi_\alpha)\}_{\alpha \in A}$ is called an atlas. If, in addition, the transition maps $\varphi_\beta \circ \varphi_\alpha^{-1} : \varphi_\alpha(U_\alpha \cap U_\beta) \to \varphi_\beta(U_\alpha \cap U_\beta)$ is smooth, then $X$ is called a *smooth or differentiable manifold*. Conceptually, Hausdorff guarantees that any sequence in $X$ converges to at most one point, locally Hilbertian implies that differential calculus can be defined locally. To further warrant that the local calculus can be smoothly extended to the entire manifold by using a smooth partition of unity following from the second countability (Lang, 1999), we also assume that the Hilbert space $\mathcal{H}$ is separable so that the norm induced by the inner product is differentiable away from 0 (Fabian et al., 2001).

Given a point $x$ in the smooth manifold $X$ and let $(U, \varphi)$ and $(V, \psi)$ be two coordinate charts centered at $x$, i.e., $x \in U \cap V$ and $\varphi(x) = \psi(x) = 0$. Pick $v \in \varphi(U) \subseteq \mathcal{H}$ and $w \in \psi(V) \subseteq \mathcal{H}$, then we say $(U, \varphi, v)$ and $(V, \varphi, w)$ are equivalent if $d(\psi \circ \varphi^{-1}) \cdot v = w$, This defines an equivalence relation, whose equivalence class is called a *tangent vector* of $X$ at $x$, and the space of all tangent vectors is called the tangent space of $X$ at $x$, denoted by $T_x X$, which is definitely a Hilbert space isomorphic to $\mathcal{H}$ (Lang, 1999). The disjoint union of all tangent spaces, $TX = \bigsqcup_{x \in X} T_x X$, is called the *tangent bundle* of $X$, which is also a smooth manifold so that the projection map $\pi : TX \to X$, given by $v_x \mapsto x$ with $v_x \in T_x M$, is smooth. Conversely, a map $f : X \to TX$ is called a *vector field* on $X$, or more generally a section of $TX$ over $X$, provided that $\pi \circ f : X \to X$ is the identity map on $X$. A curve on $X$, i.e., a map $\gamma : [a, b] \to X$, is an *integral curve* of the vector field $f$ if $f(\gamma(t))$ is tangent to $\gamma(t)$ for each $t$. If $f$ is regular enough, e.g., Lipschitz continuous, then the integral curve can be obtained by the solution of ordinary differential equation $\frac{d}{dt} x(t) = f(x(t))$. This then gives rise to the concept of the *flow* of $f$ as the map $\Phi : I \times X \to X$ such that $\Phi(t, x_0)$ is the point $x(t) \in X$ on the integral curve of $f$ passing though $x_0 = x(0)$ at $t = 0$.

Dually, we can construct the *cotangent space* $T_x^* X$ as the dual space of $T_x X$, which then gives rise to the *cotangent bundle* $T^* M = \bigsqcup_{x \in X} T_x^* X$ of $X$. A section of $T^* M$ is also called a *1-form* on $X$. From the Riemannina perspective, the primary interest is in the two-fold tensor product of the cotangent bundle: $T^* X \otimes T^* X = \bigsqcup_{x \in X} T_x X \otimes T_x X$, which is essentlly given by the disjoint union of the two-fold tensor product of the tangent space at every point on $X$. A positive definite smooth section $g$ of $T^* X \otimes T^* X$ is called a *Riemannian metric* on $X$, that is, $g_x(v, w) = g_x(v, w)$ and $g_x(v, v) \geq c\langle v, v \rangle$ for any $x \in X$, $v, w \in T_x X$, and some constant $c > 0$ independent of $x$, where $g_x$ denotes the restriction of $g$ to $T_x X \otimes T_x X$ and $\langle \cdot, \cdot \rangle$ is the inner product on $\mathcal{H}$. Note that $g_x$ essentially defines an inner product on $T_x X$ for every $x \in X$, and hence there should be no confusion to denote $g_x$ as $\langle \cdot, \cdot \rangle_x$ or simply $\langle \cdot, \cdot \rangle$ as well. In particular, given a smooth function $h : X \to \mathbb{R}$, we know that the differential $dh$ defines a 1-form on $X$, and hence evaluated at every point $x \in X$, $df_x \in T_x^* X$ is a continuous functional on $T_x X$. Then, applying the Riesz representation theorem to the Hilbert space $T_x X$ with the inner product given by the Riemannian metric, we obtain

grad $f(x) \in T_x X$ such that $df_x \cdot v = \langle \operatorname{grad} f(x), v \rangle_x$ for any $v \in T_x X$. By varying $x$, it results in a smooth vector field grad $f$, which is called the *natural gradient* of $f$.

A smooth manifold equipped with a Riemannian metric is then called a *RM*. A major advantage of a RM is the metric space structure. Following the same notation as above, given a curve $\gamma : [a, b] \to X$ on the RM $X$, its length can be defined as $L(\gamma) = \int_a^b |\dot{\gamma}(t)|dt$, where $|\dot{\gamma}(t)| = \langle \dot{\gamma}(t), \dot{\gamma}(t) \rangle^{1/2}$ with '$\cdot$' denoting the time derivative. For any $x, y \in X$, their distance is then given by $d(x, y) = \inf\{L(\gamma) : \gamma \text{ connects } x \text{ and } y\}$. If $d(x, y) = L(\gamma^*)$, then $\gamma^*$ is called a *geodesic* connecting $x$ and $y$.

In this work, we are particularly interested in space of functions $\mathcal{F}(\Omega, M)$ between two smooth manifolds $\Omega$ and $M$, where we assume $\Omega$ is compact and $M$ is a Riemannian. To construct a Riemannian structure on $\mathcal{F}(\Omega, M)$, we start with $C^\infty(\Omega, M)$, the space of smooth functions from $\Omega$ to $M$. It can be shown that $C^\infty(\Omega, M)$ is a smooth manifold modeled on $C^\infty(\Omega, TM)$ and for each $h \in C^\infty(\Omega, M)$, the tangent space is given by $T_h C^\infty(\Omega, M) = h^* TM$, the space of smooth vector fields along $h$, or equivalently, the pullback bundle of $TM$ by $h$ (Kriegl & Michor, 1997). Then, for any $v, w \in T_h C^\infty(\Omega, M)$, $g_M(v, w)$ gives a smooth real-valued function on $\Omega$, where $g_M$ denotes the Riemannian metric on $M$. Then, we can define an inner product on $T_h C^\infty(\Omega, M)$, e.g., by $\langle v, w \rangle = \int_\Omega g_M(v, w)d\text{vol}$, in which $d\text{vol}$ is a volume form on $\Omega$, e.g., it can be chosen as the pullback of the Riemannian volume form on $M$. This inner product can then be extended to a Riemannian metric on $C^\infty(\Omega, M)$. Next, if we are interested in a function $\mathcal{F}(\Omega, M)$ lager than $C^\infty(\Omega, M)$, say $L^2(\Omega, M)$, the space of square integrable functions, then we can form the tangent space $T_h L^2(\Omega, M)$ by taking the closure of $T_h C^\infty(\Omega, M)$ with respect to the topology generated by the inner product introduced above.