# OpenReview forum: "Ensemble Systems Representation for Function Learning over Manifolds"
_ICLR.cc/2024/Conference — ICLR 2024 Conference Withdrawn Submission_

### Official Review · Reviewer_o8J1 · 2023-10-30

**Soundness:** 2 fair
**Presentation:** 2 fair
**Contribution:** 1 poor
**Rating:** 1
**Confidence:** 3

**Summary:**

The paper considers the general task of function learning by dynamical ensemble systems, when on a manifold. It connects the task of function learning to ensemble dynamics and control theory, in particular (Triggiani, 1977). The main goal and the main result of the paper is not clear, nor is the comparison with existing literature. The literature review in Section 2, lists related works, but the relevance of broader literature and individual contributions is not thoroughly discussed nor explained here. Section 3 states the main definitions of controllability known in the field. In Section 4 the authors state that ensemble systems can be used for function approximation, but I believe this has been known. In Section 5 they explain that natural gradient descent can be used in ensemble systems as well. In Section 6, the authors give examples of fitting a function from a limited sample of points, using a governing ODE solving the gradient descent flow.

**Strengths:**

The paper is methodic in introducing controllability concepts. It is also an unusual approach to take to function approximation, which is not always considered in ML. The paper is written grammatically correct with good English. There does not appear to be typos of any sorts.

**Weaknesses:**

The paper is poorly written in terms of the content, the structure, and motivation of the work. It is not clear what is its main aim, nor the implications of its results. It gives very weak overview of literature. It provides very general propositions that are not directly applicable. It seems like the main result of the paper is the connection of dynamic systems used for function approximation, but this is not novel. There are very little experiments to back up the claims that this is an efficient way of function approximation.

**Questions:**

What is the main convergence result of the paper?

What are other works in the field concerned with and how does your compare with it?

Why is in Figure 3b which should show least-squares convergence a bump at time 20?

---

### Official Review · Reviewer_p7oi · 2023-10-31

**Soundness:** 2 fair
**Presentation:** 3 good
**Contribution:** 2 fair
**Rating:** 5
**Confidence:** 3

**Summary:**

This study introduces a novel framework that approaches function learning tasks through the lens of ensemble systems and control theory. The core proposition is to derive function learning algorithms by leveraging flows of continuous-time ensemble systems set on infinite-dimensional Riemannian manifolds. The research demonstrates that by adjusting the penalty strengths as control inputs, the algorithms can globally converge to any function across the manifold, ensuring ensemble controllability of these systems. The validity of the approach is further bolstered by tangible examples showcased in numerical experiments.

**Strengths:**

Quality score: 60%

To the best of my knowledge, the introduction of inhomogeneous ensemble DS on infinite-dimensional RMs and the use of flows to generate FL algorithms are technically sound. The convergence analysis is concise and rigorous, and the numerical examples provide support.

Clarity score: 90%

In general, this paper is written clearly. I have only noticed some minor points related to the references:

Reference (Weinan, 2017): Weinan is the first name, while E is the family name.

Reference (ichi Amari, 1998): It might be better to use Shun-ichi Amari or Shunichi Amari.

**Weaknesses:**

Originality score: 50%

While the introduced framework, which relies on ensemble systems on infinite-dimensional Riemannian manifolds, is indeed novel, the concept of viewing FL algorithms as discrete counterparts of continuous-time dynamic systems is not unprecedented. This work seems to extend existing concepts from control theory to describe the learning process, indicating a moderate level of technical novelty.

Significance score: 50%

Given the uncertain connections between training algorithms and phenomena associated with deep neural networks, the proposed framework may have limitations in analyzing the empirical generalization behaviors of DNNs. Therefore, the overall significance of this study might be somewhat constrained.

**Questions:**

Q1. Given the unclear relationships between training algorithms and the phenomena associated with deep neural networks, it's important to clarify how the proposed framework can be applied to analyze the empirical generalization behavior of DNNs. Could you elaborate on specific methodologies or examples where this framework can effectively contribute to understanding DNN generalization?

Q2. The empirical significance of FL on manifolds for machine learning tasks, such as image recognition, does not appear to be explicitly stated with sufficient examples in this paper. Is FL on manifolds considered a crucial technique empirically in the field of Deep Learning? If so, could you provide reasons and examples to support its significance?

---

### Official Review · Reviewer_ZBR1 · 2023-11-01

**Soundness:** 3 good
**Presentation:** 2 fair
**Contribution:** 2 fair
**Rating:** 3
**Confidence:** 3

**Summary:**

This paper introduces the concept of ensemble systems (dynamics of a set of function values) in control theory to analyze the learning process. In essence, the authors aim to model the learning process as a controlled dynamical system. While the iterative nature of the learning process naturally exhibits dynamics, this paper is particularly noteworthy because of the novel introduction of the control factor and other analogies from control theory to elucidate learning dynamics. By leveraging the theoretical behaviors of controlled ensemble systems, the authors establish certain theoretical properties of ensemble systems as a learning framework, such as convergence, from a stability perspective. Finally, the authors propose a continuous learning algorithm based on ensemble systems, which is combined with the natural gradient scheme, for optimizing learnable parameterized models. The paper also provides empirical validation through simple examples.

**Strengths:**

The introduction of control theory and ensemble systems for interpreting the learning process is intriguing to me. However, I acknowledge that my evaluation may not be rigorous due to my limited knowledge of recent developments in learning theory.

**Weaknesses:**

However, I have some concerns, mainly regarding the empirical evaluation of the proposed model.

**1. No direct comparisons with the previous method.**

In the abstract, the authors assert that they offer concrete examples that demonstrate the high efficiency and excellent generalizability of the proposed “continuous-time” algorithm compared with classical “discrete-time” algorithms. However, upon closer examination of the main paper, I struggled to locate a direct comparison between the proposed ensemble system-based learning algorithm and classical methods such as gradient descent. In fact, I was unable to find any benchmarks or comparisons with previous algorithms, which raises questions about the practical utility and persuasiveness of the proposed approach.

**2. Too simple target problems.**

The examples presented in this paper, namely the sinusoidal curve fitting and classification within a sphere, appear to be quite simplistic. While these examples are effective for illustrating the fundamental principles of the model in a visually intuitive manner, I remain somewhat skeptical about the method's performance when applied to more practical and standard problems, such as MNIST or CIFAR10.

***

Overall, I believe the experiment conducted in this paper is relatively weak, even when considering its theoretical focus. While not every paper needs to achieve SOTA results across all datasets, it is important to acknowledge that this does not excuse overly simplistic experiments. Given that this paper introduces a novel method for tackling function learning tasks, it is reasonable to expect some basic requirements and benchmark comparisons in the empirical evaluations. If the authors could include comparisons with previous methods on standard examples and demonstrate that the proposed method achieves reasonable performance, even if it does not necessarily surpass or match existing methods, I would reconsider my evaluation of this paper.

**Questions:**

- It appears that the design of the control input $u(t)$ plays a crucial role in algorithm performance. Are there any guidelines available for designing it effectively?

- Recent theoretical and empirical findings suggest that injecting stochasticity and off-manifold behavior, such as increasing the learning rate and utilizing smaller mini-batch sizes, can be beneficial for discrete gradient descent methods when compared to their continuous and smooth counterparts like gradient flow. This approach results in enhanced generalization, thanks to the implicit regularization effect [1, 2]. I would like to inquire about the authors' perspective on the concept of implicit regularization, specifically whether achieving an exact flow on the manifold truly provides an advantage for model generalization.

***

[1] Barrett, D., & Dherin, B. Implicit Gradient Regularization. In International Conference on Learning Representations 2020.

[2] Smith, S. L., Dherin, B., Barrett, D., & De, S. On the Origin of Implicit Regularization in Stochastic Gradient Descent. In International Conference on Learning Representations 2020.

---

### Official Review · Reviewer_s5Ry · 2023-11-06

**Soundness:** 3 good
**Presentation:** 2 fair
**Contribution:** 2 fair
**Rating:** 3
**Confidence:** 3

**Summary:**

This paper proposes an approach to address function learning tasks by framing them within the perspective of ensemble control problems. It establishes a connection between the stability and controllability of ensemble systems and the convergence of the function learning algorithms. Function learning over Riemannian manifolds is then considered as a natural gradient flow system, where the regularization in learning is formulated as a control for the system. To validate the benefit of the proposed framework, numerical experiments involving curve fitting and function learning over spheres are performed.

**Strengths:**

- This paper constructs an interesting and novel connection between ensemble control and function learning problems, which might facilitate other works that apply control theories to learning problems.
- The theories are established in a mathematically solid way.

**Weaknesses:**

- In control problems, desired states are typically well-defined. However, in learning tasks, desirable states that perform well on unseen test data are mostly unknown. This fundamental disparity between the two domains can make one wonder if any benefit exists from applying the proposed framework to typical learning tasks.
- Beyond achieving faster convergence to solutions minimizing training loss, what can this framework offer for a better generalization performance?
- All experiments are performed on synthetic data sets. It would be great to incorporate more practical learning problems that can obtain benefit from the proposed formulations.

**Questions:**

- Can some examples be provided that incorporate regularizers known to improve generalization, such as L2 regularization, within the proposed framework? The current ones in Section 6 make it difficult to find connections to the generalization.
- Would the control inputs considered desirable in control theory, such as minimum energy controls, still be meaningful in learning tasks according to the proposed framework? If not, what types of control inputs would be more effective?

---

### Meta-Review · Area_Chair_WHvJ · 2023-12-14

**Metareview:**

This work aims to approach function learning from a controlability standpoint. They present a formulation of continuous time learning on Riemannian manifolds, and demonstrate their method on synthetic datasets. While reviewers appreciated the conceptual approach of the paper, there were a number of aspects that were identified as weaknesses, including what the utility of this outlook was, clarity on the overall goals of the paper, the use of only simple synthetic data examples, and a lack of clarity on the novelty of this formulation relative to the literature. There was no response or discussion, and so I recommend this paper not be accepted.

**Justification For Why Not Higher Score:**

There were multiple weaknesses raised by the reviewers, including:
 - The benefit or the presented model and approach
 - clarity on the paper's overall goals
 - the limited simple synthetic data examples
 - lack of clarity on the work's novelty relative to the literature.

**Justification For Why Not Lower Score:**

N/A

---

### Decision · Program_Chairs · 2024-01-16

Reject